# *Pax3* Gene Regulated Melanin Synthesis by Tyrosinase Pathway in *Pteria penguin*

**DOI:** 10.3390/ijms19123700

**Published:** 2018-11-22

**Authors:** Feifei Yu, Bingliang Qu, Dandan Lin, Yuewen Deng, Ronglian Huang, Zhiming Zhong

**Affiliations:** Fishery College, Guangdong Ocean University, 40 East Jiefang Road, Xiashan District, Zhanjiang 524025, China; yufeifei2000@163.com (F.Y.); molgen512@126.com (D.L.); dengyw@gdou.edu.cn (Y.D.); hrl8849@163.com (R.H.); qublyuff@163.com (Z.Z.)

**Keywords:** *Pax3*, *Pteria penguin* (*Röding*, *1798*), tyrosinase, melanin, RNA interference, liquid chromatograph-tandem mass spectrometer (LC-MS/MS)

## Abstract

The paired-box 3 (*Pax3*) is a transcription factor and it plays an important part in melanin synthesis. In this study, a new *Pax3* gene was identified from *Pteria penguin* (*Röding*, *1798*) (*P. penguin*) by RACE-PCR (rapid-amplification of cDNA ends-polymerase chain reaction) and its effect on melanin synthesis was deliberated by RNA interference (RNAi). The cDNA of *PpPax3* was 2250 bp long, containing an open reading fragment of 1365 bp encoding 455 amino acids. Amino acid alignment and phylogenetic tree showed *PpPax3* shared the highest (69.2%) identity with *Pax3* of *Mizuhopecten yessoensis*. Tissue expression profile showed that *PpPax3* had the highest expression in mantle, a nacre-formation related tissue. The *PpPax3* silencing significantly inhibited the expression of *PpPax3*, *PpMitf*, *PpTyr* and *PpCdk2*, genes involved in *Tyr*-mediated melanin synthesis, but had no effect on *PpCreb2* and an increase effect on *PpBcl2*. Furthermore, the *PpPax3* knockdown obviously decreased the tyrosinase activity, the total content of eumelanin and the proportion of PDCA (pyrrole-2,3-dicarboxylic acid) in eumelanin, consistent with influence of tyrosinase (*Tyr*) knockdown. These data indicated that *PpPax3* played an important regulating role in melanin synthesis by *Tyr* pathway in *P. penguin*.

## 1. Introduction

The winged pearl oyster *Pteria penguin* (*Röding*, *1798*) (*P. penguin*) is an important marine cultured species that produces high-quality seawater pearl, whose value depends mainly on its color [1,2]. The melanin is the major pigment in *P. penguin* and largely affects the color and value of the pearl [3]. Moreover, the *P. penguin* is considered to be the best research model for melanin synthesis and color reconstruction in bivalve, because of its purely black shell in population. The mantle tissue is the main organ responsible for the formation and secretion of nacre, which is called as “mother of pearl” [4]. Inhibiting the synthesis and secretion of melanin in mantle might change the color of nacre in *P. penguin*.

Melanin plays an important role in a series of physiological processes, including pigmentation, skin photoprotection and aging [5]. In mammals, melanin synthesis is a complex process, and more than 40 genes participate in it [6,7,8]. Tyrosinase (*Tyr*) is a key rate-limiting enzyme in melanogenesis [9,10], because it catalyzes three different reactions in biosynthetic pathway of melanin [11]. The microphthalmia-associated transcription factor (*Mitf*) is a central regulator of melanogenesis, and it activates the transcription of several important genes, including *Tyr*, *Cdk2* (cyclin-dependent kinase 2) and *Bcl2* (B-cell lymphoma 2), to control melanocyte differentiation, growth and survival [6,8,12].

The paired-box 3 (*Pax3*) is a member of the paired-box family of transcription factors, and it participates in the development of central nervous system, skeletal muscles, and melanocytes [13,14]. Several studies have demonstrated that *Pax3* is frequently expressed in normal melanocytes and aggregated melanomas [15]. PAX3 directly promotes *Mitf* transcription by binding the *Mitf* promoter, functioning with SOX10 (SRY box 10) and synergizing with the CREB (cyclic-AMP responsive element-binding protein) [16]. However, at the same time, PAX3 competes with MITF by occupying the enhancer of dopachrome tautomerase (*Dct*), a downstream enzyme that functions in melanin synthesis [17,18]. Despite extensive investigations about *Pax3* being carried out over recent years, the effect of *Pax3* on melanin synthesis in bivalves is still largely speculative. 

The eumelanin, which gave organisms a brown-black color, was the main pigment in *P. penguin* [3]. Natural eumelanin is mainly composed of the monomer units 5,6-dihydroxyindole (DHI) and 5,6-dihydroxyindole-2-carboxylic acid (DHICA), with various ratios of DHI and DHICA [19]. The alkaline hydrogen peroxide oxidation of eumelanin yields pyrrole-2,3-dicarboxylic acid (PDCA) and pyrrole-2,3,5-tricarboxylic acid (PTCA) from DHI- and DHICA-derived units [5]. The PDCA and PTCA can be detected by high-performance liquid chromatography (HPLC), because they are insoluble in both acidic and alkaline solutions. Quantification of PDCA and PTCA has been extensively used to evaluate the amount and composition of eumelanin in a pigment sample using liquid chromatograph-tandem mass spectrometer (LC-MS/MS) [3,20]. 

In our previous works, we analyzed the crucial function of *Tyr* in menlanin synthesis of *P. penguin* [3]. In this study, a new *Pax3* gene from *P. penguin* was identified, and its exact function in melanin synthesis was deliberated by RNA interference (RNAi) technology. The relative genes involved in the regulation of *PpPax3* to melanin synthesis were enriched. A *Pax3*-*Tyr*-melanin axis was verified to exist in *P. penguin*.

## 2. Results

### 2.1. Cloning and Sequence Analysis of Pax3 cDNA in P. penguin

Based on the cDNA fragment of *Pax3* from transcriptome database of *P. penguin*, the complete cDNA of *PpPax3* gene was obtained by RACE-PCR. The *PpPax3* cDNA consisted of an open reading frame (452–1816) of 1365bp, a 5′-untranslated region (UTR) of 451bp, and a 3′-UTR of 434bp with a typical polyadenylation signal sequence (AATAAA) and a 31bp poly (A) tail (Figure 1). The putative amino acid sequence was 454 amino acids long. No signal peptide and transmembrane domain were found in deduced PAX3 sequence. The predicted molecular mass of *PpPax* protein was 50.7 kDa, and the theoretical isoelectric point (pI) was 8.15. The full-length cDNA sequence of *PpPax3* was submitted to Genebank with the accession no. MH558581.

### 2.2. Multiple Sequence Alignment and Phylogenetic Analysis

Amino acid sequence alignments of *Pax3* gene from *P. penguin* and other species were performed. The *PpPax3* shared the highest (69.2%) identity with *Pax3* of *Mizuhopecten yessoensis*, 52.5% with *Pax3* of *Aplysia californica*, 45.8% with *Pax3* of *Parasteatoda tepidariorum*, and 45.0% with *Pax3* of *Branchiostoma belcheri*. The deduced amino acid sequence comparison revealed a highly conserved paried box domain (PD) containing 127 amino acids and a homeodomain (HD) containing 59 amino acids (Figure 2) in *PpPax3* gene. But the octapeptide motifs located between PD and HD were not obvious in *Pax3* of *P. Pengui*, *A. californica* and *P. tepidariorum*.

The phylogenetic tree analysis was performed to indicate the evolutionary relationships of *Pax3* from different species. As shown in Figure 2B, *PpPax3* was close to *Pax3* of *M. yessoensis*, one bivalve, with a support of 89%. Three *Pax3* genes of mollusk referred, including *P. penguin*, *M. yessoensis* and *Aplysia californica*, were grouped into a close cluster. The *Pax3* of *Aplysia californica* showed high homology with that of mollusk, agreed with their taxonomic relationships. All *Pax3* genes of vertebrates referred were classified to a big clade, which exhibited farther distance to *Pax3* of invertebrates.

### 2.3. PpPax3 mRNA Expression Profile in Different Tissues

The relative expression levels of *PpPax3* gene in various tissues were compared by qRT-PCR (Quantitative real time polymerase chain reaction), with β-actin as an internal control. As shown in Figure 3, the *PpPax3* has constitutive expression in mantle, gill, adductor muscle, digestive diverticulum, foot, testis and ovary. The highest expression level of *PpPax3* was shown in the mantle, followed by the foot, without significant difference between them (*P* > 0.05). The expression of *PpPax3* in the mantle was approximately 2-fold higher than that in adductor muscle, testis and ovary with obvious significant difference (*P* < 0.05). The digestive diverticulum has the lowest expression level. Since *PpPax3* was mainly expressed in the mantle, which was responsible for nacre secretion, the mantle was selected for further studies.

### 2.4. PpPax3 Expression Was Inhibited by RNA Interference in P. penguin

To investigate the function of *Pax3* in melanin synthesis of *P. penguin*, RNA interference was performed to inhibit the expression of *PpPax3*, and qRT-PCR was employed to evaluate the silencing effects. As shown in Figure 4A, the *PpPax3* expression was reduced by 65.7% in the *Pax3*-siRNA1 group (*P* < 0.05) and 37.7% in the *Pax3*-siRNA2 group (*P* < 0.05) compared with the negative control (NC) group. This indicated that the expression of *PpPax3* was significantly knocked down by RNA interference.

### 2.5. PpPax3 Silencing Affected the Expression of PpMitf, PpTyr, PpCreb2, PpBcl2 and PpCdk2 in P. penguin

After *PpPax3* silencing, the transcripts of *PpMitf, PpTyr, PpCreb2, PpBcl2* and *PpCdk2* genes were analyzed. As shown in Figure 4B, after *PpPax3* knockdown, the transcript of *PpMitf*, a central transcription factor of melanogenesis, was significantly decreased by 64.7% in the *Pax3*-siRNA1 (Small interfering RNA) group (*P* < 0.01) and 46.7% in the *Pax3*-siRNA2 group (*P* < 0.05). The expression of *PpTyr*, the key rate-limiting enzyme for melanin synthesis, was obviously reduced by 53.3% (*P* < 0.01) by *Pax3*-siRNA1 and 33.3% (*P* < 0.05) by *Pax3*-siRNA2. The *PpCdk2* mRNA, a melanocyte growth-dependent kinase, was also depressed by 36.6% (*P* < 0.05) and 26.7% (*P* < 0.05). However, no significant difference was observed in *PpCreb2* transcript (*P* > 0.05), a regulatory factor in melanin synthesis pathway. Moreover, the transcript of *PpBcl2*, an apoptosis-related gene, was raised up to 1.9 fold through *Pax3*-siRNA1 interference (*P* < 0.05). 

### 2.6. PpPax3 Silencing Depressed Tyrosinase Activity in P. penguin

The tyrosinase activity was investigated according to the change in absorbance per minute at 475nm due to dopachrome formation from L-tyrosine of Levodopa (L-DOPA). As respected, after *PpPax3* silencing, the tyrosinase activity was obviously decreased about 48.7% in siRNA1 group (*P* < 0.05) and 31.2% in siRNA2 group (*P* < 0.05) compared to NC group (Figure 5). This indicated that the PAX3 could obviously affect the tyrosinase activity in *P. penguin*.

### 2.7. PpPax3 Silencing Decreased Melanin Content and Proportion of PDCA 

To further investigate the function of *PpPax3* in melanin synthesis in *P. penguin*, we detected the content and composition of melanin in mantle after RNA interference using LC-MS/MS. As expected, by ion spectra examination, the alkaline hydrogen peroxide oxidation products of eumelanin from *P. Penguin* were identified as PDCA and PTCA, because their mass-to-charge ratio values were 156 and 199, respectively, consistent with their molecular weight [3]. The quantity of PDCA and PTCA was calculated according to the special area of peak, which appeared at 2.39 min and 3.58 min, respectively (Figure 6A). As shown in Figure 6B, the total content of PDCA and PTCA was obviously decreased from 674.6 ng/mg to 348.4 ng/mg (by 49.3%) in *PpPax3*-siRNA1 group and to 478.5 ng/mg (by 30.1%) in *PpPax3*-siRNA2 group (*P* < 0.05). The quantity of PDCA was inhibited by 63.5% and 42.6% in *PpPax3*-siRNA1 and *PpPax3*-siRNA2 groups (*P* < 0.05). The quantity of PTCA was inhibited by 45.0% and 26.1% groups (*P* < 0.05). Moreover, after RNA interference, the proportion of PDCA in total oxidation products was obviously decreased from 19.1% to 13.5% (SiRNA1) and 14.6% (SiRNA2) (*P* < 0.05).

## 3. Discussion

*Pax* gene family encodes transcription factors that are characterized by presence of the paired box domain (PD), an octapeptide motif and homeodomain (HD) [21,22]. As reported [23], after an extensive comparison, both PD and HD of PAX3 proteins among different species were present and highly conserved, but the octapeptide might be absent, such as in Gastropoda, Annelida and in the bivalvia *Pinctada fucata*. Similarly, in this study, the obvious octapeptide only was found in vertebrates, in *B. belcheri* and one bivalve (*M. yessoensis*). No clear homologue of octapeptide could be evidenced in *P. penguin*, *A. californica* and *P. tepidariorum*.

Based on sequence homologies of PD, *Pax* gene family has been classified into four subfamilies, *Pax1/9, Pax2/5/8, Pax3/7* and *Pax4/6*. The *Pax3/7* subfamily includes *Pax3* gene and *Pax7* genes, which have high similarity in amino acid sequence. The *Pax3* and *Pax7* genes existed in the form of a ancestral gene in protostomes, ascidians and amphioxus, and then were separated into two genes in vertebrates by duplication of the ancestral gene [21]. This might be an explanation for high homology between *Pax3* of *P. penguin* and *Pax7* of other species, which implied the functional diversity of *Pax3* in *P. penguin*.

Our previous studies showed that tyrosinase was a key melanin synthase, and it played a dominant role in melanin synthesis and color formation of *P. penguin* [3]. In this report, the data showed that the knockdown of *PpPax3* caused a significant decrease in *Tyr* expression, tyrosinase activity and melanin content, similar to the influence of *Tyr* silencing [3]. The finding illustrated that PAX3 could affect the melanin synthesis by regulating the expression of *Tyr*. A *Pax3*-*Tyr*-melanin axis might exist in melanin synthesis pathway of *P. penguin*.

In humans, microphthalmia-associated transcription factor (MITF), known as a master regulator of melanogenesis, binded to the highly conserved binding motif in the regulatory region of the tyrosinase (TYR) promoter, and strongly stimulated the melanocyte-specific transcription of *Tyr* gene [7,24]. *Pax3* directly activated expression of *Mitf* and indirectly affected the expression of *Tyr* in mice [12,20]. However, compared with vertebrates, little is known about whether the MITF also involve in melanin synthesis in bivalve. Our data showed that *PpPax3* silencing significantly cut down the expression of *PpMitf* and obviously depressed the transcription of *PpTyr*. This illustrated that *Mitf* took part in the melanin synthesis of *P. penguin.* The *PpPax3* might affect *Tyr* expression and melanin synthesis through regulation to *Mitf*, similar to that in mammals. Further studies were needed to specify the existence of *Pax3-Mitf-Tyr* pathway in melanin synthesis of *P. penguin*.

As MITF is a multifunctional transcription factor that activates the transcription of various genes involved not only in melanin synthesis, but also in melanocyte proliferation and survival in mammal [12,25], we speculated that *Pax3* might also regulate melanocyte growth and survival by *Mitf* in *P. penguin.* The expressions of *PpCdk2* and *PpBcl2* were analyzed after *PpPax3* interference, because *Cdk2* and *Bcl2* respectively were important genes in melanocyte proliferation and survival in mammals, and both of them were direct MITF target genes [26,27]. In this study, the *PpPax3* silencing obviously decreased the *PpCdk2* expression, which implied *PpPax3* played an important part in control of melanocyte growth in *P. penguin*. Meanwhile, the expression of *PpBcl2* was lightly increased after *PpPax3* silencing, which implied that *PpPax3* played an important part in control of cell survival. It was worth mentioning that the *PpPax3* silencing led to an obvious decrease of *PpMitf*, but a light increase of *PpBcl2*. A possible explanation for this contradiction was that the reduction of tyrosinase protein by *PpPax3* silencing might partly damage of normal cells functions, because tyrosinase involved in several important physiological processes including pigment synthesis [9,28], innate immunity [29] and wound healing [30]. The cell damage led to the up-regulation of cell apoptosis gene *PpBcl2* through another pathway, and antagonized the depression of *PpBcl2* by *PpMitf* decrease.

In human melanocytes, PAX3 partners with SOX10 to induce melanocyte differentiation and melanin synthesis [31]. The CREB (cyclic-AMP responsive element-binding protein), as a cofactor, was inputted to PAX3 and Sox10, so that all three transcription factors induce the expression of *Mitf* [8,11,32,33]. In this report, the silencing of *Pax3* did not affect the expression of *Creb*, which indicated that CREB was not the downstream gene of *Pax3* in *P. penguin*. Further, the alone knockdown of *PpPax3* inhibited the expression of *PpMitf* and *PpTyr,* which implied that the regulation of *PpPax3* to *PpMitf* and *PpTyr* could happen independently of CREB change of in *P. penguin*. The CREB might just work as a cofactor of PAX3 to enhance the regulation effect of *PpPax3* on melanin synthesis in *P. penguin*.

RNA interference is a powerful tool and has been widely used to knock down genes to analyze the genes function, especially in human. For instance, *Pax3* SiRNA was transfected in to human metastatic melanoma to elaborate the function of *Pax3* in melanoma growth and survival [34]. Basing on these new technologies, in human, several widely-believed melanin synthesis pathways were reported, such as the cAMP(cyclic adenosine monophosphate) pathway and the Wnt (wingless-type MMTV integration site family) pathway. The cAMP pathway is a main signal pathway, which goes in the axis: MSH (melanocyte simulating hormone)-MC1R (melanocortin 1 receptor)-cAMP (cycle AMP)–PKA (protein kinase A)–CREB–MITF–tyrosinase. In Wnt pathway, PAX3, partnering with SOX10, induces the expression of *Mitf*, and then affects the expression of *Tyr*. However, in bivalve, the melanin synthesis pathway is still unclear, although some important genes involved melanin synthesis has been cloned and analyzed, such as tyrosinase fom *P. fucata* [35], *Hyriopsis cumingii* [4] and *Crassostrea gigas* [36], *Mitf* from *Meretrix petechialis* [37], *Pax3* from *P. fucata*, *M. yessoensis* and *A. californica* [23]. There has been no clear axis or pathway predicted in bivalve so far. In this research, by functional analysis, we believed that a *Pax3-Mitf-tyr* axis was existent in *P. penguin*, similarly to in humans. The PAX3, by inducing the *Mitf* expression, regulated the melanocyte differentiation, proliferation and survival. However, whether Wnt pathway existed and whether CREB worked as a member of Wnt pathway in *P. penguin* were still problems that deserved further research.

## 4. Materials and Methods 

### 4.1. Experimental Animals, RNA Isolation and cDNA Synthesis

The *Pteria penguin* (*Röding*, *1798*) samples used in this study were obtained from Weizhou Island in Beihai, Guangxi Province, China. All animals were about two years old, with shell length ranging from 12 and 15 cm. They were cultivated with the recirculating seawater at 25–26 °C for one week before the experiment. 

Total RNA from mantle (pallial zone and marginal zone), gill, adductor muscle, digestive diverticulum, foot, testis and ovary of *P. penguin* were extracted using RNeasyMini Kit (Qiagen, Gaithersburg, MD, USA), according to the manufacturer’s instructions. The integrity and quantity of RNA were detected by electrophoresis on 1% agarose gels and NanoDrop ND1000 Spectrophotometer. The cDNA was synthesized from total RNA using a Superscript II polymerase kit (TransGen, Beijing, China). 

### 4.2. cDNA Cloning and Sequence Analysis

The full-length cDNA sequence of *Pax3* was obtained with SMART RACE cDNA Amplification Kit (Clontech, Mountain View, CA, USA) and Advantage 2 cDNA Polymerase Mix (Clontech, Mountain View, CA, USA) following the manufacturer’s protocol. The nested-PCR was employed to enrich the specific DNA band. The test-PCR was used to detect the correctness of linked nucleotide sequence. All used primers were listed in Table 1.

The full-length cDNA of *Pax3* was analyzed by the BLAST program (http://www.ncbi.nlm.nih.gov/). ORF Finder (https://www.ncbi.nlm.nih.gov/orffinder/) was used to characterize the open reading fragment (ORF). Signal 4.1 (http://www.cbs.dtu.dk/services/SignalP/) and TMHMM program (http://www.cbs.dtu. dk/services/TMHMM/) were used to predict signal peptide and transmembrane. Multiple sequence alignments and phylogenetic tree were created using Clustal W and MEGA 6. The protein molecular weight and theoretical pI were analyzed using program tools (http://web.expasy.org/cgibin/protparam/protparam).

### 4.3. Quantitative Real-Time PCR (qRT-PCR) Analysis 

The qRT-PCR assays were performed using Thermo Scientific DyNAmo Flash SYBR Green qPCR Kit (Thermo scientific, Waltham, MA, USA) and the Applied Biosystems 7500/7500 Fast Real-time System (ABI, Carlsbad, CA, USA). Each sample was run in triplicate, along with the internal control gene β-actin. The specific primers were listed in Table 1. The calibration curve was established by several dilutions of standard samples, and used as a linear regression model. The 2^−∆∆*C*T^ method was applied to calculate the relative expression levels of genes.

### 4.4. RNA Interference Experiment

RNA interference was used to analyze the function of *PpPax3* gene. The *PpPax3*-siRNA1 and *PpPax3*-SiRNA2 were synthesized to specially silence the conserved domain of *PpPax3*. The GFP-SiRNA was synthesized from pEGFP-N3 plasmid, a eukaryotic-expression vector encoding green fluorescent protein, as a negative control (NC), and RNase-free water was used as a blank control (primers as Table 1). Six individuals were used in each treatment group. When RNA interference experiment, the experimental individuals were gotten out from seawater and dried in air until the shells were slightly open. Then, 100 μL SiRNAs (small interfering RNA) at a final concentration of 1 μg/μL were gently injected into adductor muscle of experimental individuals, which was then put into seawater and cultivated for 3 days in the lab to have a recovery [38]. At the 4th day, the experimental individuals were injected with same dose of SiRNAs again, and had another recovery for 3 days. At the 7th day, the experimental animals were killed and the tissues were collected for RNA extraction, tyrosinase activity assay and melanin analysis. 

### 4.5. Tyrosinase Activity Assays

Tyrosinase activity assays were performed as described previously with minor modification [19,39]. Briefly, 1g mantle was homogenized in 1 mL of 0.1 mol/L phosphate buffer (pH 6.8) and centrifugated to obtain the tissue supernate. The 0.5mL of 5 mmol/L L-DOPA (3,4-dihydroxyphenylalanine) was mixed with 2.4 mL of 0.1 mmol/L PBS (phosphate buffer saline), followed by the addition of 0.5 mL of tissue supernate. The mixture was incubated at 37 °C for 30 min, and then the absorbance of the mixture was measured at 475 nm. The total tryosinase activity of every group was represented by the change of absorbance value in 30 min. One unit (U) of the tyrosinase activity was defined as increased or decreased absorbancy per minute at 475 nm. The relative tyrosinase activity was shown using the percentage of every group in NC group.

### 4.6. Isolation and Oxidation of Total Melanin

The total melanin from mantle of *P. penguin* was isolated and oxidized following our previous report [3]. Briefly, 1 g mantle sample was finely homogenized and incubated in 15 mL phosphate buffer (pH 7.4) with 2% (m/V) papain at 55 °C for 20 h. The mixture was centrifuged at 10,000 rpm for 10min to obtain the precipitate, which was successively washed with 2 mL mineral ether for 3 times, 2 mL ethanol for 3 times and 2 mL water 3 times. Then, the black precipitate was dried and measured as raw melanin production. 

The raw melanin production was dissolved in 8.6 mL of 1 mol/L K_2_CO_3_ and 0.8 mL of 30% H_2_O_2_. The mixture was heated under reflux at 100 °C for 20 min. After cooling, 0.4 mL of 10% Na_2_SO_3_ was added to end the reaction. The mixture was acidified to pH 1.0 with 5 mL of 6 mol/L HCl and then was extracted twice with 70 mL of ether. The supernanant was collected and dried to obtain crystalline residue, which was finally redissolved in mobile phase and filtered by 0.45 μm organic membrane for liquid chromatograph-tandem mass spectrometer (LC-MS/MS) analysis. 

### 4.7. LC-MS/MS Assay of Melanin 

The content and component of melanin were detected by LC-MS/MS according to previous description [40] with some modification. The chromatographic separation was performed using an Acquity ultraperformance liquid chromatography (UPLC) system (Waters, Milford, MA, USA) consisting of a Waters ACQUITY UPLC HSS T3 (2.1 × 50 mm, 1.7 μm particle size). The mobile phase A and B was 0.1% (*v*/*v*) of formic acid in deionized water and 0.1% (*v*/*v*) of formic acid in methanol, respectively. The ratio of mobile phase A in total mobile phase was gradually decreased from 90% to 0% within 5 min. The cycle time was 5 min per injection. Analyses were performed at 40 °C at a flow rate of 0.3 mL/min. MS/MS detection was performed using a Xevo TQ triple quadrupole mass spectrometer operated in positive electrospray ionization (ESI) mode similar to Yu et al. [3]. The source temperature and desolvation temperature were 150 °C and 550 °C, respectively. The cone gas flow, desolvation gas flow and collision gas flow were 50 L/h, 1100 L/h and 0.14 mL/min (argon), respectively. The analytes were monitored in multireaction monitoring mode (MRM). Specific parameters were given as Table 2.

### 4.8. Statistical Analysis

ANOVA analysis was performed using SPSS 19.0 (IBM, Armonk, NY, USA) to detect the significance of difference among different samples. Significant difference was indicated by * (*P* < 0.05), highly significant difference was indicated by ** (*P* < 0.01) and extremely significant difference was indicated by *** (*P* < 0.001).

## 5. Conclusions

In this study, we characterized a new *Pax3* gene from *P. penguin*. Tissue expression profile showed that *PpPax3* had the highest expression in mantle, a nacre-formation related tissue. The *PpPax3* silencing significantly inhibited the transcription of *PpPax3, PpMitf, PpTyr* and *PpCdk2*, genes involved in *Tyr*-mediated melanin synthesis, but had no effect on *PpCreb2* and an increase effect on *PpBcl2*. Furthermore, the *PpPax3* silencing obviously decreased the tyrosinase activity, the total content of eumelanin and the proportion of PDCA in eumelanin, similar to the influence of *Tyr* silencing. Thus, we believed that *PpPax3* played an important role in melanin synthesis by indirectly regulating the expression of *Tyr* in *P. penguin*. The *Pax3*-*Tyr*-melanin axis was considered a potential strategy in melanin synthesis of *P. penguin*.

## Figures and Tables

**Figure 1 ijms-19-03700-f001:**
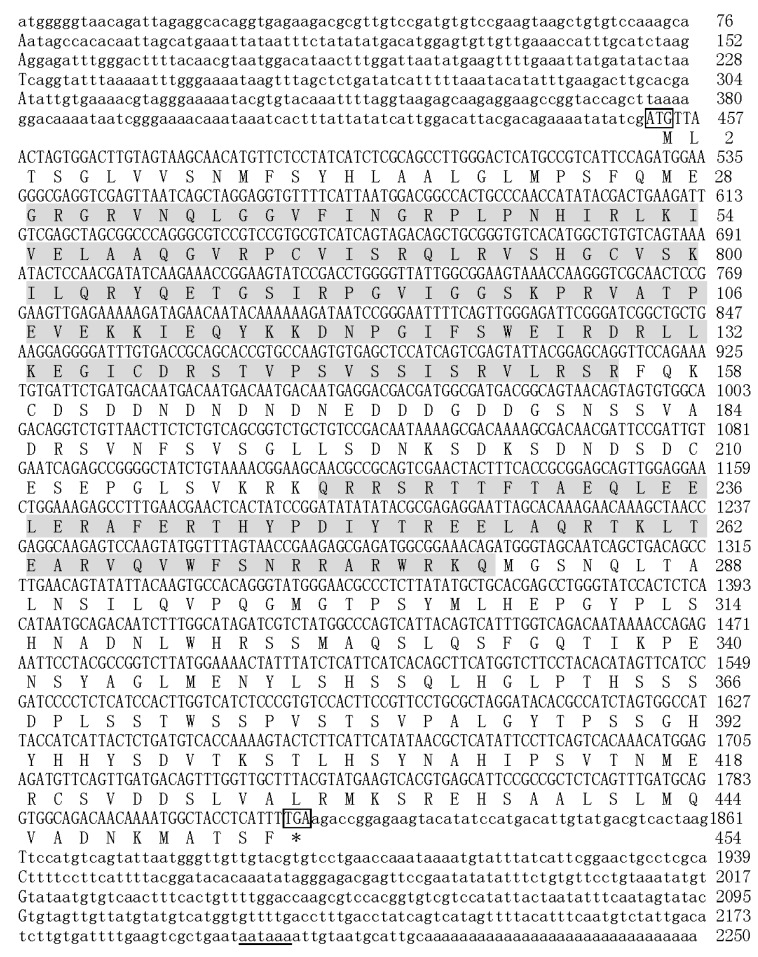
Nucleotide and deduced amino acid sequences of *PpPax3*. PD (paried box domain) and HD (homeodomain) were shown in grey boxes. The ORF (open reading frame) and deduced amino acid sequences were shown in uppercase. The 5′-UTR (untranslated region) and 3’-UTR were shown in lowercase. The initiation codon (ATG) and the stop codon (TGA) were boxed. The putative polyadenylation signal (aataaa) was underlined.

**Figure 2 ijms-19-03700-f002:**
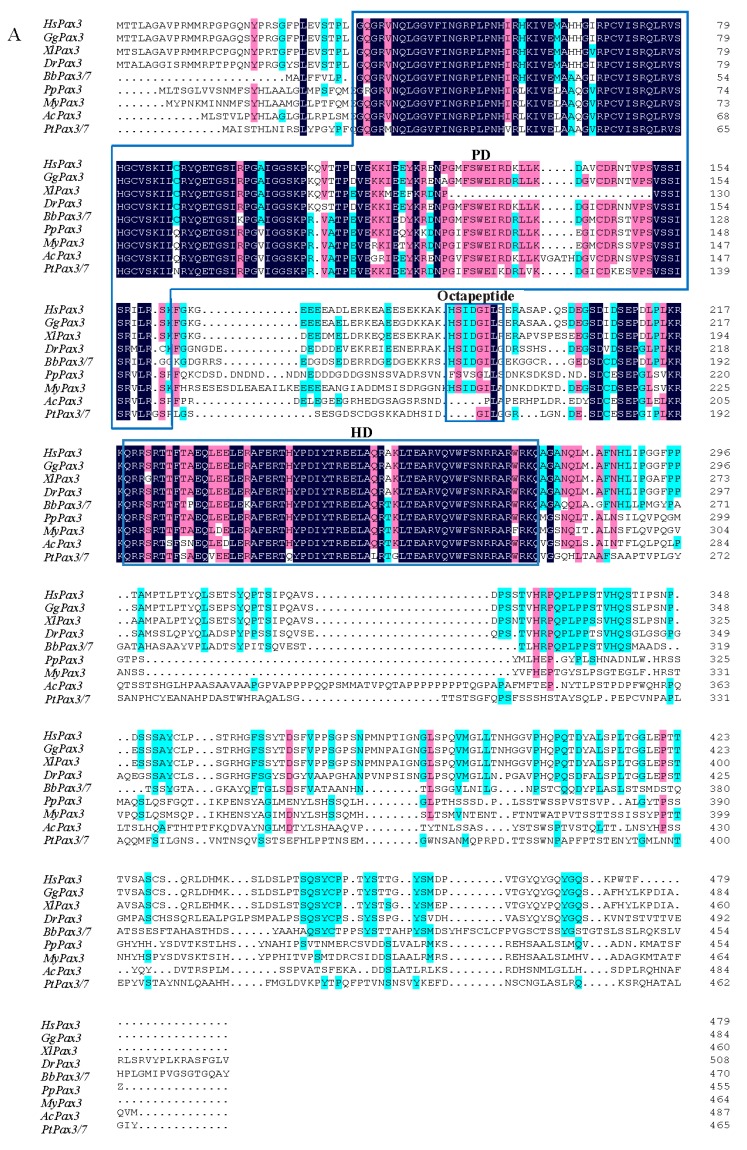
Sequence alignment and phylogenetic analysis. (**A**) Multiple sequence comparison of *Pax3* among different species including *Pteria penguin*(*Röding*, *1798*) (*PpPax3*, MH558581), *Mizuhopecten yessoensis* (*MyPax3*, XP_021364914.1), *Aplysia californica* (*AcPax3*, XP_012943435.1), *Parasteatoda tepidariorum* (*PtPax3/7*, BBD75270.1), *Branchiostoma belcheri* (*BbPax3/7*, ABK54280.1), *Danio rerio* (*DrPax3*, AAC41253.1), *Homo sapiens* (*HsPax3*, NP_852122.1), *Xenopus laevis* (*XlPax3*, AAI08574.1) and *Gallus gallus* (*GgPax3*, BAB85652.1). The conserved amino acids were written in black background, and similar amino acids were shaded in green and pink. PD, octapeptide motif and HD were indicated in blue boxes. (**B**) Phylogenetic tree of *Pax3* genes. Numbers in the branches represented the bootstrap values (as a percentage). ▲ meaned the *Pax3* of *P. penguin*.

**Figure 3 ijms-19-03700-f003:**
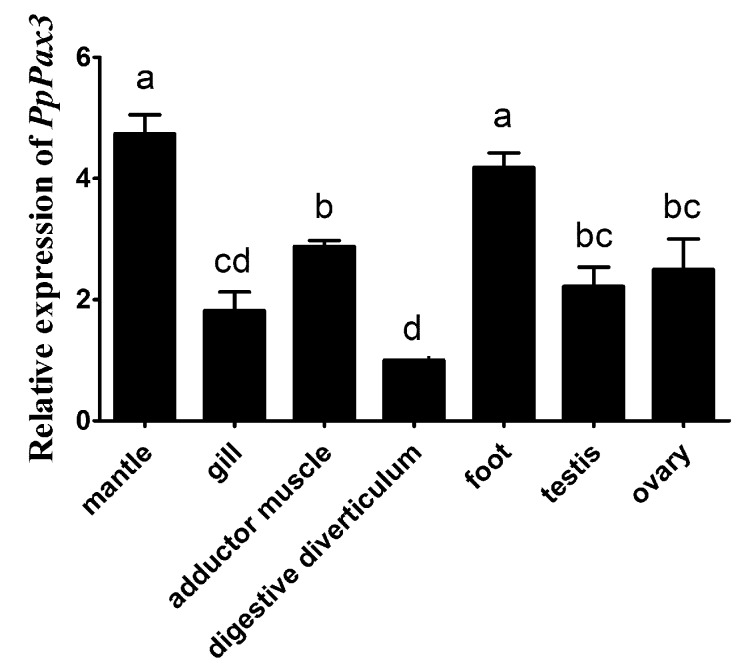
Relative expression of *PpPax3* in various tissues of *P. penguin* estimated by qRT-PCR. Each bar was a mean of 6 pearl oysters. Error bars were the SD. Different letters (a, b, c and d) meaned significant difference (*P* < 0.05).

**Figure 4 ijms-19-03700-f004:**
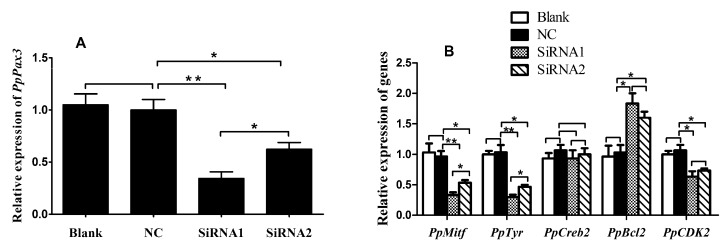
Expression of *PpPax3, PpMitf, PpTyr, PpCreb2, PpBcl2* and *PpCdk2* after *PpPax3* RNA interference (RNAi). (**A**) Expression of *PpPax3*. (**B**) Expression of *PpMitf, PpTyr, PpCreb2, PpBcl2* and *PpCdk2*. The qRT-PCR was done with RNA samples from blank group (RNase-free water), NC group (GFP-siRNA), *PpPax3*-siRNA1 group and *PpPax3*-siRNA2 group. The β-actin of *P. penguin* was used as an internal control. Each bar was a mean of 6 individuals. Significant difference was indicated by * (*P* < 0.05) and highly significant difference was indicated by ** (*P* < 0.01).

**Figure 5 ijms-19-03700-f005:**
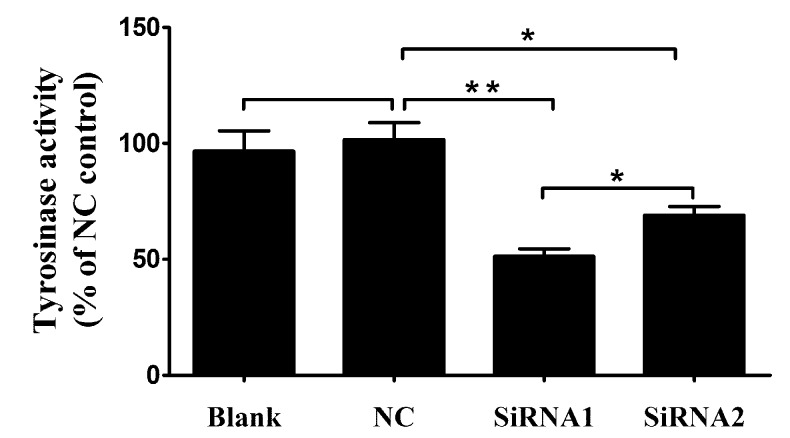
The tyrosinase activity of the control groups and RNA interference groups. The tyrosinase activity was performed with samples from blank group (RNase-free water), NC group (green fluorescent protein (GFP)-siRNA), *PpPax3*-siRNA1 group and *PpPax3*-siRNA2 group. The tyrosinase activity was shown with percentage of every group and NC. Each bar was a mean of 6 pearl oysters. Significant difference was indicated by * (*P* < 0.05) and highly significant difference was indicated by ** (*P* < 0.01).

**Figure 6 ijms-19-03700-f006:**
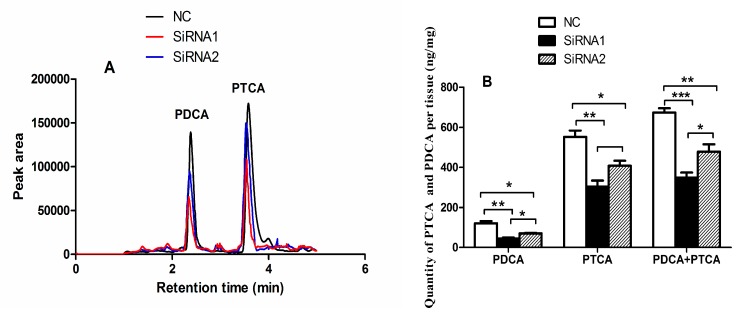
LC-MS/MS analysis of oxidation products of melanin in *P. penguin*. (**A**) HPLC (High Performance Liquid Chromatography) chromatograms of oxidation products of melanin from NC group and RNAi groups. (**B**) The content of PDCA and PTCA from NC group and RNAi groups. Each bar was a mean of 6 pearl oysters. * meaned significant difference (*P* < 0.05); ** meaned very significant difference (*P* < 0.01); *** meaned extremely significant difference (*P* < 0.001).

**Table 1 ijms-19-03700-t001:** Primers used in the study.

Primer	Sequence (5′–3′)	Application
*PpPax3*-outer-F	GGACGGCCACTGCCCAACCATATACG	3′RACE
*PpPax3*-inner-F	AAGTAAACCAAGGGTCGCAACTCCG	nest-3′RACE
*PpPax3*-outer-R	GGAACCTGCTCCGTAATACTCGACTGATGG	5′RACE
*PpPax3*-inner-R	TCAGCAGCCGATCCCGAATCT	nest-5′RACE
UPM (Universal Primer)	TAATACGACTCACTATAGGGCAAGCAGTGGTATCAACGCAGAGT	RACE universal primer
NUP (Nested Universal Primer)	AAGCAGTGGTATCAACGCAGAGT	Nest-RACE universal primer
*PpPax3*-test-F	GAATGCTCCGTAAACGTTATTG	cDNA test
*PpPax3*-test-R	GACAACAAAATGGCTACCTCAT	cDNA test
*PpPax3*-siRNA1-F	GCGTAATACGACTCACTATAGGGGTAAACCAAGGGTCGCAAC	RNAi
*PpPax3*-siRNA1-R	GCGTAATACGACTCACTATAGGGCGTTGTCGCTTTTGTCGCT	RNAi
*PpPax3*-siRNA2-F	GCGTAATACGACTCACTATAGGGGATAATCCGGGAATTTTCAGTTGGG	RNAi
*PpPax3*-siRNA2-R	GCGTAATACGACTCACTATAGGGGATAGTGAGTTCGTTCAAAGGCTCT	RNAi
GFP-siRNA-F	GATCACTAATACGACTCACTATAGGGATGGTGAGCAAGGGCGAGGA	RNAi
GFP-siRNA-R	GATCACTAATACGACTCACTATAGGGTTACTTGTACAGCTCGTCCA	RNAi
*PpPax3*-qPCR-F	TCCGTGCGTCATCAGTAGAC	qRT-PCR
*PpPax3*-qPCR-R	CCCTTGGTTTACTTCCGCCA	qRT-PCR
*PpTyr*-qPCR-F	CTCAGGGAAGGGATCAGCTT	qRT-PCR
*PpTyr*-qPCR-R	AGACCCTCTGCCATTACCAA	qRT-PCR
*PpMitf*-qPCR-F	TGTTACCTAAATCTGTTGATCCAG	qRT-PCR
*PpMitf*-qPCR-R	AAATTAGCTGGACAGGAAGAGGAG	qRT-PCR
*PpCreb2*-qPCR-F	AACTCCCAGTGAAGCAGACA	qRT-PCR
*PpCreb2*-qPCR-R	GCTCCCCAACAGTAGCCAAT	qRT-PCR
*PpBcl2*-qPCR-F	TGAGGCACAGTTCCAGGATT	qRT-PCR
*PpBcl2*-qPCR-R	ACTCTCCACACACCGTACAG	qRT-PCR
*PpCdk2*-qPCR-F	TGGATTTGCTCGGACACTTG	qRT-PCR
*PpCdk2*-qPCR-R	TCTACTGCCCTGCCATACTT	qRT-PCR
β-actin-F	CGGTACCACCATGTTCTCAG	qRT-PCR
β-actin-R	GACCGGATTCATCGTATTCC	qRT-PCR

**Table 2 ijms-19-03700-t002:** Details of mass spectrometric detection.

Compand	Parent Ion (m/z)	Product Ion (m/z)	Conc Voltage (V)	Collision Energy (eV)	Retention Time (min)
PDCA	155.98	138.01	30	8	2.39
PTCA	199.99	182.09	30	8	3.58

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
