# Peer review of "Pax3 Gene Regulated Melanin Synthesis by Tyrosinase Pathway in Pteria penguin"

_ijms, 2018, doi:10.3390/ijms19123700_

Round 1

Reviewer 1 Report

The manuscript titled “Pax3 gene regulated melanin synthesis by tyrosinase pathway in Pteria Penguin” is well written and documents that PpPax3 plays an important regulating role in melanin synthesis by Tyr pathway in P. penguin. Before this manuscript can be accepted, some modifications are required.

Minor comments:

1.       Language corrections need to be done

2.       Pteria Penguin needs to be written as Pteria penguin throughout the manuscript

3.       Please enlarge Fig 6A to show clearly the differences between NC vs siRNAs used

Major comments

1.       Expand the methods to indicate how the RNAi experiments were done. How were the oysters treated and the delivery of the siRNA molecules or plasmids was done is not clear

2.       No visual representation of showing color variation of the treated vs control is shown

3.       Methods are incomplete in most sections

4.       Comparisons of the melanin biosynthesis pathways between the bivalve and humans need to be clarifies and elaborated

5.       Discussion of the applicability of siRNA targeting Pax3 in humans and demonstration in human melanocytes would add significant weight to this manuscript

Author Response

Dear reviewer:

Thank you for the suggestion about the manuscript “Pax3 gene regulated melanin synthesis by tyrosinase pathway in Pteria Penguin”. According to your suggestions, the manuscript has been revised as follows:

Minor comments

1.       Language corrections have been done throughout the manuscript in the new version.

2.      The “Pteria Penguin in whole paper has been instead by Pteria penguinor  “Pteria penguin (Röding, 1798) ” . The “P. Penguin” has also been instead by “P. penguin” .

3.      The Fig 6A has been enlarged to show clearly the differences between NC and siRNAs samples.

Major comments

1.      We have expanded the 4.4 section of experiment methods and tried to indicate how RNAi experiments were done, including how to open the shells and how the SiRNA were injected into the adductor muscle of experimental individuals.

2.      Indeed, we didn’t obtain visual light-colored individual in this experiment. After 7 days of RNA interference, we observed obvious changes in expressions of melanin-related genes, tyrosinase activity and melanin content in mantle, but no visual change in shell color. A possible explanation was that there was massive melanin pre-existing in shell, which was purely deep black, so the decrease of melanin content was not displayed on the shell color. In our experiment, we also tried to raise the frequency and time of RNA interference, but the high-intensity Pax3 silencing could affect the viability and survival rate of P. penguin, maybe because Pax3 and its downstream genes participated in several physiological processes, such as pigment synthesis, innate immunity, wound healing and development.

3.      The experiment methods have been complemented or modified in 4.2-4.7 sections. An extra 4.8 section was added into Methods to introduce the statistical method.

4.        Comparisons of the melanin biosynthesis pathways between the bivalve and humans have been added into the Discussion as a separate paragraph (the last paragraph of Discussion).

5.      The applicability of SiRNA targeting Pax3 in humans and the function of Pax3 in melanin synthesis of human have been added into the last paragraph of Discussion.

Reviewer 2 Report

This is an interesting paper, which adds up to the general knowledge on melanogenesis and its evolution in animals, and which possesses a practical value. There are several questions and suggestions on improving the text, and some minor technical errors.

1.      The title does not reflect fully the content of the paper which emphasizes also differences and relationship between various taxons (as to Pax 3 genes).

2.      Melanin assay: 7 days from application of siRNA seems too short to observe changes in melanization of mantle on the level of melanin content. Was there any pre-existing melanin? What is the kinetics of producing nacre, compared to this period? What is the mechanism of pigmentation of the shell (transfer of melanin from the mantle to the nacre/shell)? If the method of melanin degradation was applied, was there an opportunity to check also pheomelanin content?

3.      In humans and other mammals (as the process of melanogenesis in P. penguin is in the paper compared to the process in mammals) the main signal pathway goes in the axis: MSH - Mc1r (receptor for MSH) – cAMP – PKA – CEBP – MiTF - tyrosinase. Pax 3 accompanies the induction of MiTF but acts independently on MC1R, so it is, indeed, aside the main pathway. Is there any analogy (or homology?) between the two networks of regulation – in mammals and in Mollusks? Is there an equivalent of signaling via MC1R in Mollusks?  Compare Ortonne and Ballotti, J Dermatol Treat 2000 (see below). 

4.      Please supplement “Material and methods” with the statistics applied (as a sub-chapter).

5.      Citations – Please supplement according to the remarks: Slominski et al., Physiol Rev. 2004, PMID:15383650 (e.g. in line 37-38), Vachtenheim & Borovansky, Exp Dermatol 2010, PMID:20201954 (here also on MITF relation to SOX10 and PAX3, and to the expression of other melanogenesis-related factors (e.g. in line 39-41); also: D’Mello et al, IJMS 2016, PMID:27428965 (e.g. in line 49), and Donoghue et al., BioEssays 2008, PMID:18478530 (e.g. line 43-49 somewhere – this paper concerns particularly the engagement of various genes to the embryogenesis of neural crest which is the precursor of melanocytes in Vertebrates, including Pax3, from the evolutionary point of view – so important for these divagations); Ortonne %& Ballotti, J Dermatol Treat 2000; https://doi.org/10.1080/09546630050517621, this paper and Slominski et al. 2004 show the main path of the signal transduction towards melanogenesis with PKA and cAMP (e.g. to cite in Conclusions or earlier in the part of Discussion where the aside position of regulation via Pax 3 and CREB/CEBP is discussed).

6.      Please maintain the correct form of the scientific nomenclature of species: Pteria penguin (Röding, 1798), or P. penguin. The second part of the name is not capitalized, both parts in Italics. Please correct carefully through the text. Please also add at least in Material and methods the author’s phrase (Röding, 1798).

7.      Please have English edited. There are some errors and typos, please check carefully. E.g.: l. 11 “and it plays…”, l. 39-40 “and it activates…”, l. 60 –“melanin”, l. 251 – tyrosinase is NOT a synthetase (E.C.6), but an oxidoreductase (E.C.1). Instead; “a key enzyme in melanogenesis”, or, at least “a key melanin synthase”; l. 310 – “in Table 1”, l. 367 “a nacre formation-related tissue”, l. 478: “Pigment Cell Melanoma Res”, and, perhaps, others.

Author Response

Dear reviewer:

Thank you for the suggestion about the manuscript “Pax3 gene regulated melanin synthesis by tyrosinase pathway in Pteria Penguin”. According to your suggestions, the manuscript has been revised as follows:

1.      It is a bit difficult to point the difference and relationship of PpPax3 between various taxons in title. Because both the definite Pax3-Tyr-melanin axis and the predicted Pax3-Mitf-Tyr-melanin axis in P. penguin were similar to mammals, although the function of Creb and the relationship of Creb and Pax3 might be different among P. penguin and other species. So if I could, we preferred to indicate the differences and relationship of PpPax3 between various taxons in text, especially in Discussion.

2.      The samples were collected after 7 days of RNA interference according to some reports. Sun et al. (2018) detected the melanin content of human epidermal melanobytes at day1, day2 and day3 after PMEL RNA interference, and found that melanin content was obviously reduced at day2 and further declined at day3. So in our research, the first SiRNA reference lasted 3 days, the second SiRNA reference lasted another 3 days to enhance the silencing effect, and experimental samples were collected at the 7th day. Actually, in previous research (Yu et al. 2018), we found 7 days was enough to observe the changes in melanin content of the mantle, but not enough for the change in shell color. A possible explanation was that there was massive melanin pre-existing in shell, which was purely deep black, so the decrease of melanin content was not displayed on the shell color.

The shell of the bivalve oyster consists of two mineralized layers including an inner nacreous and an outer prismatic layer. Melanin was also secreted from the epithelial cells of the mantle and transferred from the mantle to the nacre layer and the shell, to construct the color of the nacre and shell.

The eumelanin produces brown-black color, and the pheomelanin produces red-yellow color. The shell color of P. penguin was purely deep black, and eumelanin was found to be the main pigment in P. penguin (Yu et al. 2018). So we speculated the change of eumelanin content was the main pigment change, and the decrease of pheomelanin content might fail to be detected in P. penguin, because of its very low background concentration. Nevertheless, the check of pheomelanin content will be beneficial to understand the melanin synthesis pathway in next work.

3.      In mammals, the cAMP pathway is a main signal pathway, which goes in the axis: MSH-MC1R- cAMP– PKA – CREB – MiTF – tyrosinase. In Wnt pathway, PAX3, partnering with SOX10, induces the expression of Mitf, and the action of PAX3 was independently on MC1R. However, in Mollusks, the melanin synthesis pathway is still unclear, although some important genes involved melanin synthesis has been cloned and analyzed, such as tyrosinase in Pinctada fucata, Hyriopsis cumingii and Crassostrea gigas, Mitf in Meretrix petechialis, Pax3 in Gastropoda, Pinctada fucata, Mizuhopecten yessoensis and Aplysia californica. No clear axis or pathway was predicted in mollusks, maybe because there were no enough genes were cloned and analyzed in one species. According to our research, we believed that a Pax3-Mitf-tyr axis was existent in P. penguin, which was similar to human. The PAX3, by inducing the Mitf expression, regulated the melanocyte differentiation, proliferation and survival. However, it is doubt whether cAMP pathway via MC1R exists in mollusk, because we couldn’t clone the MC1R gene from P. penguin, and no Mc1R gene of bivalve had been cloned and studied, as we known.

4.      The “Statistical analysis” has been added into the 4.8 subsection of Materials and Methods, as a sub-chapter.

5.      We thank the reviewer for the suggestion to cite some articles, because they were classical and in-depth. The 5 classical articles have been cited in relative position of paper and listed in Reference.

6.      The “Pteria Penguinin whole paper has been instead by “Pteria penguin” (Röding, 1798) or “P. penguin”.

7.      Some grammatical errors and typos have been modified. E.g. l. 11 “and it plays…”, l. 39-40 “and it activates…”, l. 42-43 “and it participates”, l. 60 –“melanin”, l. 351 “and it played”, l. 351 “tyrosinase was a key melanin synthase”, l. 456 and l. 468 “in Table 1”, l. 17 and l. 606 “a nacre formation-related tissue”, l. 742 “Pigment Cell Res”, and so on.

References

Sun, L.; Hu, L.; Zhang, P.; Li, H.; Sun, J.; Wang H.; Xie X.; Hu, J. Silencing of PMEL attenuates melanization via activating lysosomes and degradation of tyrosinase by lysosomes. Biochem Biophys Res Commun. 2018, 503, 2536-2542. doi: 10.1016/j.bbrc.2018.07.012.

Round 2

Reviewer 1 Report

The authors have addressed all the concerns. This manuscript is now acceptable for publication.